# Clinical Utility of Ghrelin-O-Acyltransferase (GOAT) Enzyme as a Diagnostic Tool and Potential Therapeutic Target in Prostate Cancer

**DOI:** 10.3390/jcm8122056

**Published:** 2019-11-22

**Authors:** Juan M. Jiménez-Vacas, Enrique Gómez-Gómez, Antonio J. Montero-Hidalgo, Vicente Herrero-Aguayo, Fernando L-López, Rafael Sánchez-Sánchez, Ipek Guler, Ana Blanca, María José Méndez-Vidal, Julia Carrasco, José Lopez-Miranda, María J. Requena-Tapia, Justo P. Castaño, Manuel D. Gahete, Raúl M. Luque

**Affiliations:** 1Maimonides Institute for Biomedical Research of Córdoba (IMIBIC), 14004 Córdoba, Spain; b12jivaj@uco.es (J.M.J.-V.); enriquegomezgomez@yahoo.es (E.G.-G.); antonio.montero.96@gmail.com (A.J.M.-H.); b22heagv@uco.es (V.H.-A.); ferll_@hotmail.com (F.L.-L.); patologiahrs@gmail.com (R.S.-S.); ipek.guler@imibic.org (I.G.); anblape78@hotmail.com (A.B.); mjosemv@yahoo.es (M.J.M.-V.); julia.carrasco.sspa@juntadeandalucia.es (J.C.); md1lomij@uco.es (J.L.-M.); josefa.requena.sspa@juntadeandalucia.es (M.J.R.-T.); justo@uco.es (J.P.C.); bc2gaorm@uco.es (M.D.G.); 2Department of Cell Biology, Physiology, and Immunology, University of Córdoba, 14071 Córdoba, Spain; 3Hospital Universitario Reina Sofía (HURS), 14004 Córdoba, Spain; 4Centro de Investigación Biomédica en Red de Fisiopatología de la Obesidad y Nutrición, (CIBERobn), 14004 Córdoba, Spain; 5Urology Service, HURS/IMIBIC, 14004 Córdoba, Spain; 6Anatomical Pathology Service, HURS, 14004 Córdoba, Spain; 7Department of Innovation and Methodology, IMIBIC, 14004 Córdoba, Spain; 8Oncology Department, IMIBIC, 14004 Córdoba, Spain; 9Lipids and Atherosclerosis Unit, Reina Sofia University Hospital, 14004 Córdoba, Spain

**Keywords:** GOAT-enzyme, prostate cancer, diagnosis, therapy, PSA

## Abstract

Recent data suggested that plasma Ghrelin O-Acyl Transferase enzyme (GOAT) levels could represent a new diagnostic biomarker for prostate cancer (PCa). In this study, we aimed to explore the diagnostic and prognostic/aggressiveness capacity of GOAT in urine, as well as to interrogate its putative pathophysiological role in PCa. We analysed urine/plasma levels of GOAT in a cohort of 993 patients. In vitro (i.e., cell-proliferation) and in vivo (tumor-growth in a xenograft-model) approaches were performed in response to the modulation of GOAT expression/activity in PCa cells. Our results demonstrate that plasma and urine GOAT levels were significantly elevated in PCa patients compared to controls. Remarkably, GOAT significantly outperformed PSA in the diagnosis of PCa and significant PCa in patients with PSA levels ranging from 3 to 10 ng/mL (the so-called PSA grey-zone). Additionally, urine GOAT levels were associated to clinical (e.g., Gleason-score, PSA levels) and molecular (e.g., *CDK2*/*CDK6*/*CDKN2A* expression) aggressiveness parameters. Indeed, *GOAT* overexpression increased, while its silencing/blockade decreased cell-proliferation in PCa cells. Moreover, xenograft tumors derived from GOAT-overexpressing PCa (DU145) cells were significantly higher than those derived from the mock-overexpressing cells. Altogether, our results demonstrate that GOAT could be used as a diagnostic and aggressiveness marker in urine and a therapeutic target in PCa.

## 1. Introduction

Prostate cancer (PCa) is one of the tumor pathologies with the highest incidence among the male population and represents a severe health problem worldwide [1]. Early detection of PCa is a crucial step for the successful management of patients with this pathology. The detection of PCa cases has been refined with the establishment of the prostatic specific antigen (PSA) as the gold standard tool for PCa diagnosis. However, although PSA levels are used in clinical practice, this diagnostic tool poses important limitations, especially in the so-called “grey zone” (defined as a PSA range of 3–10 ng/mL). The most important drawback is the compromised specificity, which is due to the fact that there are several non-tumor factors (i.e., benign prostatic hyperplasia, prostatitis) associated to an increase of PSA levels [2]. For these reasons, the anatomo-pathological analysis of prostate biopsies, which represent a highly invasive technique, is still essential to appropriately diagnose PCa. Therefore, the use of PSA levels as diagnostic tool for PCa is linked to many unnecessary biopsies, which are potentially associated to clinical side effects (e.g., infections, bleeding), overdiagnosis and overtreatment, thus leading to an increase in economic burden [3]. In this sense, many efforts have been made in order to identify novel and more accurate PCa biomarkers, which has led to the identification of certain promising candidates [4]. However, these candidates have not been globally introduced in the clinical practice, likely due to the lack of validation in sufficiently ample cohorts and/or the high cost associated to their determination. For these reasons, new biomarkers for PCa diagnosis are necessary, ideally non-invasive biomarkers with also prognostic/aggressiveness and/or therapeutic potential.

In this regard, a ghrelin system has emerged as a pivotal regulatory axis in PCa pathophysiology [5,6], as well as a source of potential PCa biomarkers, since certain peptides derived from this pleiotropic system (i.e., native ghrelin, In1-ghrelin splice variant) are secreted by PCa cells and associated to PCa aggressiveness [7,8]. Ghrelin axis is controlled, at least in part, by the Ghrelin O-Acyl Transferase (GOAT or MBOAT4), an enzyme involved in the acylation and, thus, activation of ghrelin. This acylation is necessary for the binding of ghrelin to GHSR1a [9]. The GOAT enzyme is mainly produced in stomach and pancreas [10] and regulated by energy status and relevant metabolic cues [11]. Interestingly, our laboratory has previously reported that GOAT is overexpressed in PCa tissues and released by PCa cells [12]. Moreover, we have recently demonstrated in a cohort of 312 patients that plasma GOAT levels are higher in patients with PCa and particularly, in patients with clinically significant PCa (SigPCa, defined as PCa with Gleason score ≥ 7) as compared to controls, which included patients with suspected PCa but negative biopsy results and those with indolent PCa (defined as PCa with Gleason 6) [13]. Indeed, this study also showed that plasma GOAT levels were able to outperform the diagnostic capability of PSA, especially in the grey zone [13]. However, to date, the clinical utility of GOAT as a diagnostic (or prognostic/aggressiveness) tool in urine samples (a body sample less invasive and with a more prostate-specific content) from PCa patients has not been explored. Furthermore, although the actions of other elements of the ghrelin system (ghrelin, In1-ghrelin) in PCa have been well defined [7,8], the putative pathophysiological role of GOAT in this cancer type remains unknown. For these reasons, in this study, we aimed to explore, for the first time, the clinical utility of urine GOAT levels (and compared with plasma levels) to diagnose PCa, using an ample cohort of patients (almost 1000 patients) and to determine the role of GOAT in the pathophysiology of PCa.

## 2. Experimental Section

### 2.1. Patients and Samples

This study was approved by the Hospital Ethic Committee and written informed consent from all patients was obtained. All samples were obtained through the Andalusian Biobank (Nodo Córdoba, Servicio Andaluz de Salud, Spain). The patients included in this study were divided in 3 different cohorts.

Cohort 1:healthy volunteers (*n* = 97) that donated urine and blood samples.Cohort 2:Patients with suspect of PCa but negative results in the biopsy (*n* = 549).Cohort 3:Patients diagnosed with PCa (biopsy-proven, *n* = 347). Specifically, this cohort was divided in patients with non-significant PCa (NonSigPCa; defined as Gleason score of 6 in the biopsy; *n* = 143; cohort 3a), and in patients with significant PCa (SigPCa; defined as Gleason score ≥ 7 on the biopsy; *n* = 204; cohort 3b).

This is a retrospective study wherein patients (both from cohorts 2 and 3) were collected between 2013 and 2015 by consecutive recruitment of individuals with suspicion of PCa that underwent a transrectal ultrasound (TRUS) guided prostate biopsy according to clinical practice in the Urology Department of Reina Sofia Hospital (Córdoba, Spain). Blood and plasma samples were collected early in the morning after an overnight fast and just before the prostate biopsy. Recommendations for biopsy indication were suspicious findings on digital rectal examination (DRE), PSA > 10 ng/mL, or PSA 3–10 ng/mL if free PSA ratio was low (usually, <25–30%), and in patients with previous biopsies, a persistent suspicion of PCa (i.e., persistently elevated PSA, suspicious DRE, etc.). For transrectal prostate biopsy, 12 biopsy cores were obtained from patients undergoing the first biopsy procedure and a minimum of 16 biopsy cores for those who had a previous biopsy. All biopsy specimens were analyzed by experienced urologic pathologists according to the International Society of Urological Pathology 2005 modified criteria [14]. Tumor regions (*n* = 84) were identified from the Formalin-Fixed Paraffin-Embedded (FFPE) samples by expert urologic pathologists as previously reported [15,16] and used to isolate RNA and perform gene expression analyses. The FFPE pieces were taken from radical prostatectomies (patients belonging to cohort 3).

### 2.2. GOAT and PSA Determinations

A commercial ELISA (MBS2019923; MyBioSource, San Diego, CA, USA) was used to determine urine and plasma GOAT levels following the instructions of the manufacturer. The ELISA kit shows a detection limit lower than 0.31 ng/mL and a detection range of 0.78–50 ng/mL, as well as an intra- and inter-assay accuracy with a coefficient of variation lower than 10% and 12%, respectively. The donated urine samples were stored in 1.5 mL aliquots at −80 °C. Urine samples were diluted 1:100 before performing the assay. Measurement of PSA levels was performed in the laboratory service of the Reina Sofia University Hospital of Córdoba using the Chemiluminescent Microparticle Immunoassays technology (7k70; Abbott, Madrid, Spain) following the manufacturer’s instructions.

### 2.3. Cell Culture and Reagents

DU145 and LNCaP cell lines were obtained from American Type Culture Collection (ATCC; Manassas, VA, USA), cultured according to the manufacturer’s instructions, validated by analysis of short tandem repeats sequences using GenePrint 10 System (Promega, Barcelona, Spain) and checked for mycoplasma contamination by PCR as previously reported [8,16]. DU145 cells were selected for functional in vitro and in vivo analyses based on its high expression levels of In1-ghrelin, the main oncogenic element of the ghrelin axis in prostate cancer, which is also a putative target of GOAT [8]. The GOAT inhibitor “GO-CoA-Tat” (032-37; Phoenix Biotech, Burlingame, CA, USA) was resuspended in water and used at 10 µM since this dose has been previously reported to be effective reducing GOAT activity [17].

### 2.4. Transient Transfection with siRNAs

For silencing assays, 200,000 cells (DU145) were seeded in 6-well culture plates and grown until 70% confluence was achieved. Then, cells were transfected with a specific siRNA against GOAT (s54791; Thermo Fisher Scientific, Madrid, Spain) or with the control siRNA (“scramble”; 4390844; Thermo Fisher Scientific) at 100 nM using Lipofectamine-RNAiMAX (Thermo Fisher Scientific) according to the manufacturer’s instructions. The efficiency of *GOAT* silencing was confirmed by qPCR and subsequently, proliferation assay was performed in siRNA-transfected or scramble-transfected cells.

### 2.5. Stable Transfection with Plasmids

A DU145 cell line was stably transfected with pCDNA3.1 vector containing *GOAT* transcript (OHu31938; GenScript, Leiden, Netherlands) and with a “mock” or empty-control pCDNA3.1 vector (GenScript, Leiden, Netherlands) following the manufacturer’s instructions. The transfected cells were selected adding Geneticine (Thermo Fisher Scientific) at 1% to the culture media. The efficiency of *GOAT* overexpression was confirmed by qPCR and subsequently, proliferation assay was performed in response to transfected (GOAT-overexpression vs. mock) cells. Moreover, in vivo xenograft analyses were performed using these transfected (GOAT-overexpression vs. mock) cells (see below).

### 2.6. Cell Proliferation

Cell proliferation was evaluated using Alamar-Blue assay (Bio-Source International, Camarillo, CA, USA) in DU145 cells, as previously reported [16]. Briefly, cells were seeded in 96-well culture plates at a density of 3000–5000 cells/well and serum-starved for 24 h. Then, fluorescence (560 nm) was evaluated using the FlexStation III system (Molecular Devices, Sunnyvale, CA, USA) after 3 h of incubation with Alamar-Blue compound at 10%. This assay was performed during 3 days in response to different experimental conditions (silencing, overexpression, or treatment).

### 2.7. RNA Extraction and Retrotranscription

Total RNA from FFPE samples was isolated and DNase-treated using the Maxwell 16 LEVRNA FFPE Kit (Promega, Madison, WI, USA) in the Maxwell MDx 16 Instrument (Promega, Madrid, Spain) according to the manufacturer’s instructions. Additionally, total RNA from PCa cell lines was extracted using TRIzol Reagent (Thermo Fisher Scientific), followed by DNase treatment using RNase-Free DNase Kit (Qiagen, Hilden, Germany). Total RNA concentration and purity were assessed using Nanodrop One Spectrophotometer (Thermo Fisher Scientific). Total RNA was retrotranscribed using random hexamer primers and the cDNA First Strand Synthesis kit (Thermo Fisher Scientific).

### 2.8. Real-Time qPCR

The expression levels of selected transcripts were evaluated by real-time qPCR (RT-qPCR) using a Mx3000P thermocycler (Agilent, Madrid, Spain). Development and validation of the primers used in the RT-qPCR have been previously reported by our laboratory [18,19,20]. Specific primers used in this study were previously validated [12]: *GOAT* (sense: TTGCTCTTTTTCCCTGCTCTC; antisense: ACTGCCACGTTTAGGCATTCT; 161 bp); *ACTB* (sense: ACTCTTCCAGCCTTCCTTCCT antisense: CAGTGATCTCCTTCTGCATCCT; 176 bp); *GAPDH* (sense: AATCCCATCACCATCTTCCA; antisense: AAATGAGCCCCAGCCTTC; 122 bp). To control for variations in the efficiency of the retro-transcription reaction, mRNA copy numbers of the different transcripts analyzed were adjusted by a normalization factor, which was calculated with the expression levels of *ACTB* and *GAPDH* using GeNorm 3.3 (CMMG, Ghent, Belgium) [21].

### 2.9. Xenograft Assay

Experiments with mice were carried out according to the European Regulations for Animal Care under the approval of the university/regional government research ethics committees. Ten-week-old male athymic BALB/cAnNRj-Foxn1nu mice (*n* = 5; Janvier Labs, Le Genest-Saint-Isle, France) were subcutaneously grafted in one of the flanks with 106 mock-transfected (*n* = 5 tumors) and in the other flank with 106 stably GOAT-transfected DU145 cells (*n* = 5 tumors), which were resuspended in 100 mL basement membrane extract (Trevigen, Gaithersburg, MD, USA). Tumor growth was monitored once per week for one month using a digital caliper. After euthanasia of mice, each tumor was dissected, fixed, and sectioned for RNA isolation (snap-frozen) and histopathological-examination after hematoxylin-eosin staining. The examination of the number of mitosis and positive KI67 cells in the immunohistochemistry sections of xenograft tumors were performed by expert anatomo-pathologists as previously reported [16].

### 2.10. Statistical Analyses

Statistical differences between two groups (cell lines and patients’ data) were calculated by unpaired parametric t-test or a nonparametric Mann–Whitney *U* test, according to normality, assessed by the Kolmogorov–Smirnov test. For differences among the three groups, a one-way ANOVA analysis was performed. For cohorts’ clinical descriptive characteristics comparison, a chi-square test and ANOVA test with Bonferroni post hoc analysis were performed. Spearman’s or Pearson’s bivariate correlations were performed for quantitative variables according to normality. Statistical analyses were assessed using GraphPad Prism 7 (GraphPad Software, La Jolla, CA, USA) or SPSS version 17.0. (IBM, Armonk, NY, USA). All the in vitro experiments were performed at least 3 independent times (*n* ≥ 3) and with at least 2 technical replicates. Univariate logistic regressions were performed for the estimation of the effect of urine GOAT and PSA levels on the risk of SigPCa and NonSigPCa to further assess the predictive capacity of each biomarker separately by using receiver operating characteristics (ROC) analysis with area under the curve (AUC). A multivariate logistic regression was performed to detect risk factors for SigPCa and NonSigPCa, and the confidence intervals (CIs) of odds ratios (ORs) were calculated by ordinary bootstrapping techniques with 2000 boostrap replicates. Logistic regression models and ROC curve analysis were assessed using R software (version 3.5.0.; The R Foundation, Vienna, Austria). Statistical significance was considered when *p* < 0.05. A trend for significance was indicated when *p* values ranged between >0.05 and <0.1.

## 3. Results

### 3.1. Description of the Cohort

The clinical characteristics of the three different cohorts evaluated (*n* = 993 patients) are depicted in Table 1. Patients with PCa (cohort 3) were older compared to patients with negative biopsy (cohort 2) and healthy patients (cohort 1) (67 years (62–72) vs. 63 years (57–69) vs. 62 years (57–67), respectively; *p* < 0.01). Patients with PCa had significantly higher plasma PSA levels compared to healthy patients (cohort 3 vs. cohort 1: 6.64 (4.49–11.32) vs 0.82 (0.57–1.33) ng/mL; *p* < 0.05), while a similar, albeit non-significant trend was found compared to patients with negative biopsy (cohort 3 vs. cohort 2: 6.64 (4.49–11.32) vs. 5.27 (3.84–7.39) ng/mL; *p* = 0.11). No differences in BMI between groups were found. The proportion of patients with previous biopsy and normal DRE was significantly higher in patients with negative biopsy compared to the patients with PCa (cohort 2 vs. cohort 3; *p* < 0.01). The percentage of patients with family history of PCa did not differ between patients with PCa and with negative biopsy. Finally, 59% of the patients with PCa (cohort 3) had a Gleason score (GS) of 7 or higher on the biopsy (those classified as SigPCa; cohort 3b; *n* = 204 and 8.8% (*n* = 18) presented metastasis at the diagnosis (Table 1). Patients with SigPCa (cohort 3b) had significantly higher plasma PSA levels compared to healthy patients and negative biopsy patients (*p* < 0.001), while a similar, albeit non-significant difference was found compared to patients with NonSigPCa (cohort 3a).

### 3.2. Levels of GOAT in Non-Invasive Samples from Patients with and without PCa

Consistently with a previous study [13], the plasma levels of GOAT were significantly higher in SigPCa patients compared to healthy individuals, NegBiopsy- and NonSigPCa-patients (Appendix A). Similarly, urine levels of GOAT were found to be higher in SigPCa patients as compared to healthy individuals and NegBiopsy patients (Appendix A). Moreover, urine GOAT levels from NonSigPCa patients were higher compared to healthy individuals and NegBiopsy patients (Appendix A). Remarkably, although the ROC curve analysis revealed that the levels of both plasma and urine GOAT were able to significantly discriminate healthy individuals vs. PC patients, healthy individuals vs. SigPCa patients, NegBiopsy vs. PCa patients and NegBiopsy vs. SigPCa patients, urine GOAT significantly outperformed the capability of plasma GOAT to distinguish among these groups of patients (Appendix A). Therefore, based on these results, we decided to continue the present study by focusing on the analysis of GOAT urine levels.

### 3.3. Comparison of Diagnostic Capacity of Urine GOAT and Plasma PSA

In order to investigate the diagnostic capacity of urine GOAT levels and compare it with that of plasma PSA levels, we evaluated these levels in patients with initial suspect of PCa (cohorts 2 and 3). In particular, urine GOAT levels and plasma PSA levels were significantly higher in PCa patients compared to NegBiopsy patients (Figure 1a,b). No differences were found when comparing the AUCs of GOAT and PSA to detect PCa (Figure 1c). Similarly, urine GOAT levels and plasma PSA levels were significantly higher in SigPCa patients compared to NegBiopsy + NonSigPCa patients (Figure 1d,e), while no differences were found when comparing the AUCs of GOAT and PSA to detect SigPCa (Figure 1f). However, when patients in the grey zone were analyzed, although both urine GOAT and plasma PSA levels were significantly higher in patients with PCa compared to those with NegBiopsy (Figure 1g,h), only urine GOAT levels were able to significantly discriminate between PCa vs. NegBiopsy patients (Figure 1i). In addition, urine GOAT levels (but not PSA levels) were higher in SigPCa compared to NegBiopsy +NonSigPCa patients (Figure 1j,k) and were able to distinguish between SigPCa patients and NegBiopsy +NonSigPCa patients (Figure 1l).

In addition, to address the potential clinical utility of the measurement of both proteins (GOAT and PSA), we performed a decision curve analysis on our data, as proposed by Vickers and Elkin [20]. Based on the net plotting against the threshold probabilities for the comparisons between GOAT and PSA estimates, a clear benefit of GOAT against PSA was found, particularly in the mid-(0.2–0.5) range of the risk thresholds for PCa (Figure 1m) and SigPCa (Figure 1n).

Relevantly, although other PCa diagnostic markers can be altered in response to inflammatory conditions of the prostate, our results show a mild but non-statistically significant increase of urine GOAT levels in patients with prostatic inflammation as compared to healthy individuals (Appendix A).

We next applied a multivariate analysis adjusted with variables usually considered in clinical practice for the diagnose of PCa (PSA, age, DRE, etc.; Table 2, Table 3, Table 4 and Table 5) to evaluate the association of urine GOAT levels with the presence of PCa and SigPCa. This analysis revealed that urine GOAT levels were independently associated with an increased risk of PCa and SigPCa in the full cohort and in the cohort of patients in the grey zone of PSA. Additionally, urine GOAT levels were also found to be independently associated with an increased risk of PCa and SigPCa, even when PSA levels were excluded from the analyses.

### 3.4. Correlation of Urine GOAT Levels with Clinical and Molecular Parameters of Tumor Aggressiveness

The urine levels of GOAT were positively correlated with age (Figure 2a), PSA levels (Figure 2b), Gleason score (Figure 2c), plasma C-reactive protein (CRP) levels (Figure 2d) and negatively correlated with plasma Apolipoprotein A1 (APOA) levels (Figure 2e) in PCa patients. In addition, GOAT urine levels were also associated to clinical parameters of aggressiveness. Specifically, a positive correlation between urine GOAT levels and the expression levels (in the tumor pieces) of *CDK6*, *EGF*, *EZH2* and *NF-KB* was found in patients with PCa (Figure 2f–h). Moreover, *CDK2* and *SIRT1* expression in the tumor pieces of these patients tended to directly, while *CDKN2A* inversely, be correlated with GOAT urine levels (Figure 2i–l).

### 3.5. In Vitro and In Vivo Effects of the Modulation of GOAT Expression and/or Activity

The overexpression of *GOAT* (Appendix A) increased the proliferation rate of DU145 cells at 48 and 72 h (Figure 3a). Similar results were observed in one experiment using LNCaP cells (Appendix A). On the other hand, *GOAT* silencing (Appendix A) decreased the proliferation rate of DU145 cells at 24, 48, and 72 h (Figure 3b). Moreover, the inhibition of GOAT activity (using the specific GOAT inhibitor GO-CoA-Tat) evoked a significant decrease in the proliferation rate of DU145 cells after 24 h of incubation (Figure 3c). Moreover, xenograft tumors derived from DU145 cells overexpressing *GOAT* (Appendix A) were significantly higher (Figure 3d), showed a greater number of mitosis per mm2 (Figure 3e) and a higher KI67 staining (Figure 3f) than the tumors derived from control cells (transfected with the mock plasmid).

## 4. Discussion

The measurement of plasma PSA levels remains the current gold standard to diagnose PCa, which represents one of the tumors types with the highest incidence worldwide [1]. Unfortunately, PSA continues to show important limitations (especially in the range of 3–10 ng/mL, also named the “grey zone”), including compromised specificity, inasmuch as non-tumoral conditions (e.g., infections, inflammation) can also increase PSA levels [2]. Consequently, numerous unnecessary biopsies are carried out in the current clinical practice, generating preventable adverse effects to patients and increasing public costs [3]. Therefore, considerable research efforts have been focused on the identification of novel biomarkers that could complement or even replace plasma PSA in order to improve the capacity of the clinicians to diagnose PCa. In this sense, our group has recently demonstrated that GOAT, an enzyme involved in the acylation and activation of ghrelin and, likely, other components of this hormonal axis (e.g., In1-ghrelin, In2c-ghrelin) [9], are overexpressed in PCa tissues, but most importantly, secreted by PCa cells [12]. Interestingly, results previously reported by our group suggested that plasma GOAT levels might be used as a putative complement for plasma PSA (especially in the grey zone of PSA) for the diagnosis of PCa [13]. In the present study, we further corroborated in an ampler cohort of patients (almost 1000 patients) that GOAT plasma levels are higher in patients with PCa (especially those diagnosed with clinically significant PCa) as compared to those with negative biopsy, which is consistent with previous results reported by our group analyzing a more limited cohort of patients (*n* = 312) [13]. Moreover, we demonstrated herein that GOAT levels can also be detected in urine, being also significantly increased in PCa patients compared to control individuals. Remarkably, the differences found in urine GOAT levels were significantly more pronounced among control and cancer groups than those obtained when analyzing plasma GOAT levels from the same patients. This observation could be due to the fact that urine is enriched in prostate-derived proteins compared to the plasma and therefore, urine can act as a more reliable and specific non-invasive fluid for the diagnosis of PCa patients [22].

In line with this idea, the results obtained herein demonstrate that high urine GOAT levels were associated to a higher risk of developing PCa as well as clinically significant PCa, independently of other parameters clinically relevant for PCa diagnosis and PCa patient’s management, including Gleason score, PSA, age and DRE [14,23,24]. Indeed, the diagnostic capacity of urine GOAT levels to identify patients with PCa is comparable to that of the current gold standard plasma PSA levels. However, it should be noted that the diagnostic capacity of urine GOAT levels significantly outperforms the capacity of plasma PSA to detect PCa patients in the PSA grey zone (wherein the capacity of PSA is significantly worse). In fact, when analyzing patients in the grey zone of PSA, urine GOAT levels (but not PSA levels) are able to discriminate between SigPCa patients and patients with negative result in the biopsy or with NonSigPCa. Therefore, although no direct comparison between GOAT and other emerging and well-studied biomarkers [25,26] have been performed (inasmuch as different cohorts and approximations have been used), these results suggest that GOAT could represent a novel and valuable complement for the PSA as non-invasive diagnostic biomarker to diagnose PCa (and/or SigPCa), especially in patients with PSA ranging from 3 to 10 ng/mL. Indeed, the results presented herein are merely based on the determination of urine GOAT, and it can be therefore anticipated that the development of novel diagnostic tools that integrate urine GOAT levels with other laboratory and/or clinical parameters could result in enhanced diagnostic potential. In line with this, additional prospective validations to assess the potential of diagnostic models that include GOAT in combination with other available parameters are warranted.

Remarkably, we also found that urine GOAT levels were directly correlated to key clinical parameters associated to PCa aggressiveness, such as Gleason score [14], age [27] and PCR [28]. Henceforth, GOAT urine levels could be also used as non-invasive biomarker for the PCa aggressiveness status. This idea was reinforced by the fact that the urine levels of GOAT were also associated to the expression levels of *CDK6*, *CDKN2A*, *EZH2*, *SIRT1* in PCa tissues, all of them being closely associated to PCa aggressiveness [29,30,31,32].

This study is also the first to provide novel evidence supporting the pathological role of GOAT in PCa development and progression. Specifically, we observed that overexpression of *GOAT* evoked an increase of tumor aggressiveness features (i.e., cell proliferation) in vitro and in vivo. Specifically, GOAT-overexpressing DU145 cells induced bigger tumors compared to mock-overexpressing DU145 cells. Consistently with this, GOAT-overexpressing tumors showed a higher number of mitotic cells and a higher percentage of KI67-positive cells than the mock-control tumors. Therefore, this report demonstrates that GOAT plays an pathophysiological role in PCa, which, together with other elements of the ghrelin system that we have recently demonstrated that also have oncogenic potential and are activated by the GOAT enzyme (i.e., ghrelin and In1-ghrelin splicing variant) [8], might integrate a regulatory circuit that is altered in patients with PCa to promote the oncogenic capacity of PCa cells. Notably, the silencing of *GOAT* expression resulted in a decrease of PCa cells proliferation, reinforcing the oncogenic role of GOAT in PCa. In line with this, we tested, for the first time in tumor cells, the GO-CoA-Tat compound, which has been previously reported to specifically inhibit GOAT activity [17]. Consistently with the silencing results, the blockade of GOAT activity reduced the aggressiveness of PCa cells in vitro, further supporting the oncogenic role of GOAT and suggesting the potential novel therapeutic role of GOAT in PCa.

The present study has some limitations. First, despite the prospectively collected information, it followed a retrospective design. Secondly, the use of TRUS biopsy for PCa diagnosis, although it remains as the standard in most populations, suffers from random error compared with template biopsy [33] and therefore, could affect the prediction results. Furthermore, at the time of patients’ recruitment, mpMRI was not established in the clinical practice of our institution and therefore, a further prospective validation study in an additional cohort collected following current recommendation of performing mpMRI previous to the biopsy (GUIAS 2019) will be designed. This study was carried out with plasma and urine samples, while the ELISA kit manufacturer recommends using serum instead of plasma (although the recovery using EDTA-plasma samples is higher compared to that obtained when using serum samples; 96% vs. 89%, respectively)”.

Taken together, our results demonstrate that (1) GOAT plays an oncogenic role in PCa, (2) urine GOAT levels are directly associated to key clinical parameters of aggressiveness, and (3) urine GOAT levels outperform, at least in the grey zone, the capacity of plasma PSA to distinguish between SigPCa/PCa patients and non-PCa patients. Therefore, this report demonstrates that the GOAT enzyme could represent a novel diagnostic and aggressiveness biomarker and a potential and effective therapeutic target in PCa.

## Figures and Tables

**Figure 1 jcm-08-02056-f001:**
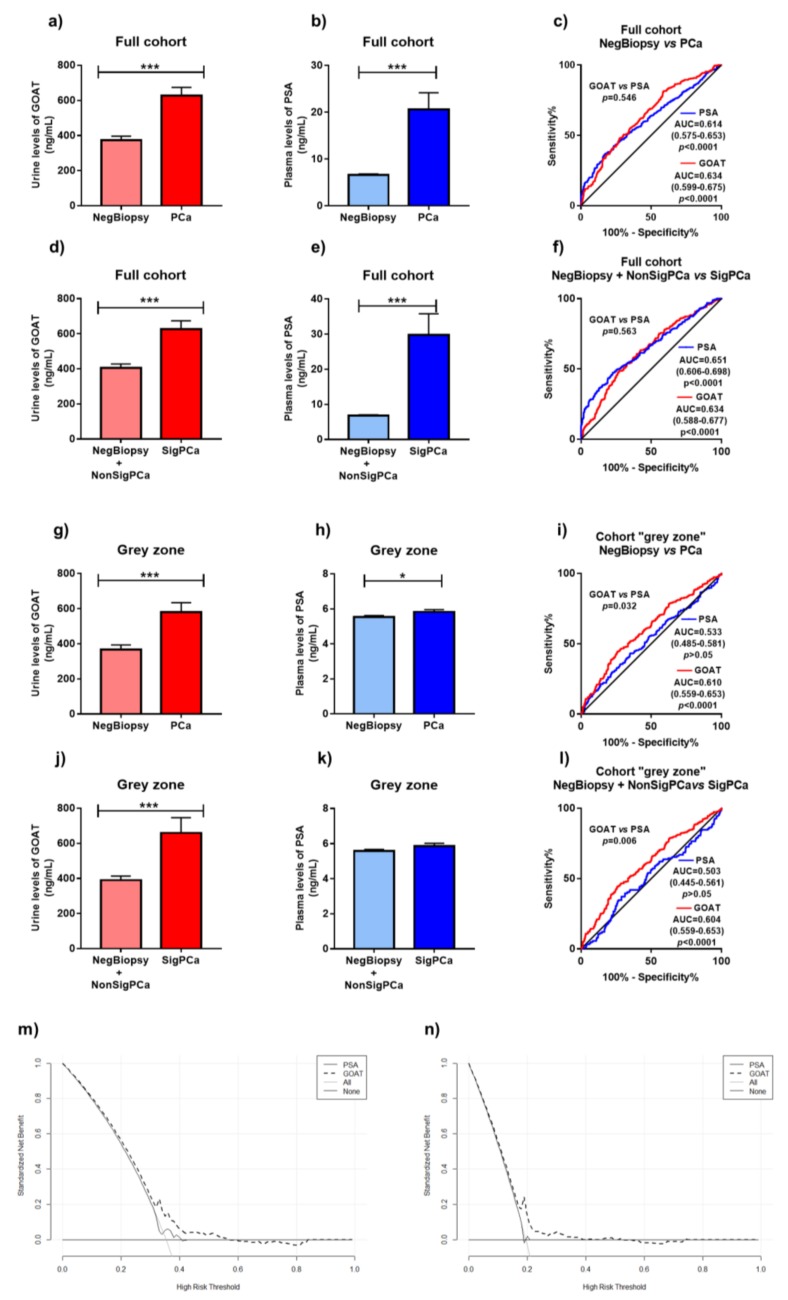
Urine Ghrelin O-Acyl Transferase enzyme (GOAT) and plasma PSA levels according to patient categorization. (**a**,**b**) Comparison between urine GOAT (**a**) and plasma PSA (**b**) levels in patients with suspect of PCa but with negative results in the biopsy (NegBiopsy; *n* = 549) and patients diagnosed with PCa (PCa; *n* = 347). (**c**) Comparison between the receiver operating characteristic (ROC) curves analyses of the capacity of GOAT (**red line**) and PSA (**blue line**) to discriminate among NegBiopsy and PCa patients. (**d**,**e**) Comparison of urine GOAT (**d**) and plasma PSA (**e**) levels in NegBiopsy patients (*n* = 549) and patients diagnosed with non-significant PCa (NonSigPCa; *n* = 143) vs. patients diagnosed with significant PCa (SigPCa; *n* = 204). (**f**) Comparison between the ROC curves analyses of the capacity of GOAT (**red line**) and PSA (**blue line**) to discriminate among NegBiopsy and NonSigPCa patients vs. PCa patients. (**g**,**h**) Comparison between urine GOAT (**g**) and plasma PSA (**h**) levels in patients in the grey zone of PSA (range 3–10 ng/mL) with suspect of PCa but with negative result in the biopsy (NegBiopsy; *n* = 411) and patients in the grey zone of PSA diagnosed with PCa (PCa; *n* = 225). (**i**) Comparison between the ROC curves analyses of the capacity of GOAT (**red line**) and PSA (**blue line**) to discriminate among NegBiopsy and PCa patients in the grey zone of PSA. (**j,k**) Comparison of urine GOAT (**j**) and plasma PSA (**k**) levels of patients in the grey zone of PSA with NegBiopsy and patients in the PSA grey zone diagnosed with NonSigPCa vs. PCa patients in the PSA grey zone diagnosed with SigPCa (*n* = 124). (**l**) Comparison between the ROC curves analyses of the capacity of GOAT (**red line**) and PSA (**blue line**) to discriminate among patients in the grey zone of PSA with NegBiopsy and those diagnosed with NonSigPCa vs. patients in the PSA grey zone diagnosed with SigPCa. (**m**,**n**) Results of the decision curve analysis. The net benefit for the prediction of PCa (**m**) and SigPCa (**n**) on biopsy is shown, by using the different models (GOAT and PSA) as a function of the risk threshold, compared to the benefits of strategies for treating all patients (**grey thin line**) and treating none (**grey thick line**). In all cases, data represent mean ± SEM. Asterisks (*, *p* < 0.05; ***, *p* < 0.001) indicate values that significantly differ between groups.

**Figure 2 jcm-08-02056-f002:**
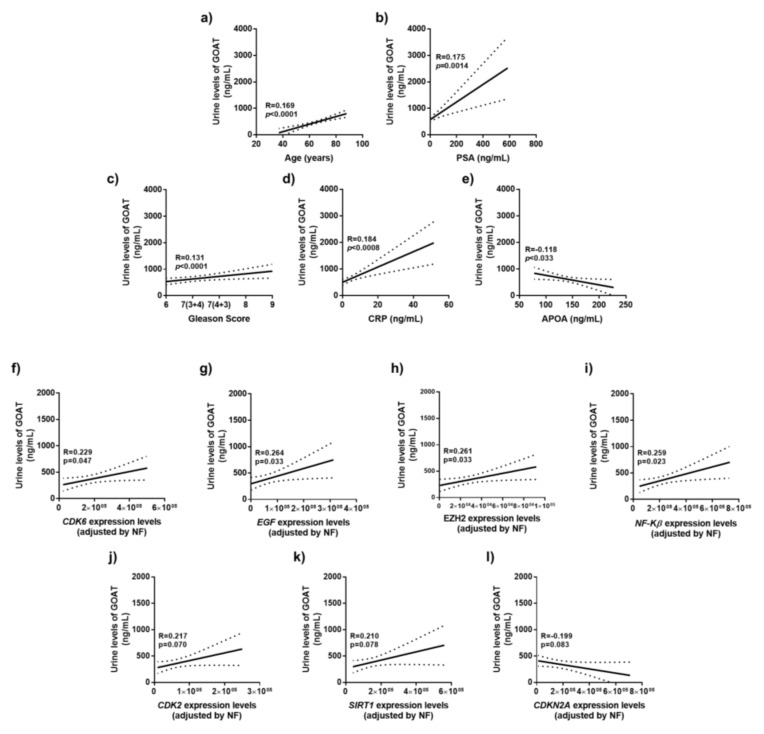
Correlations of urine GOAT levels and molecular parameters. Correlations of urine GOAT levels in PCa patients with the tissue expression levels of *CDK6*, *EGF*, *EZH2*, *NF-KB*, *CDK2*, *SIRT1* and *CDKN2A*. mRNA levels were determined by qPCR and adjusted by a normalization factor (calculated with the expression levels of *ACTB* and *GAPDH* using GeNorm). Coefficients of correlation (R) were evaluated by Pearson’s test. The graphics show the lineal adjusted method and mean confidence interval.

**Figure 3 jcm-08-02056-f003:**
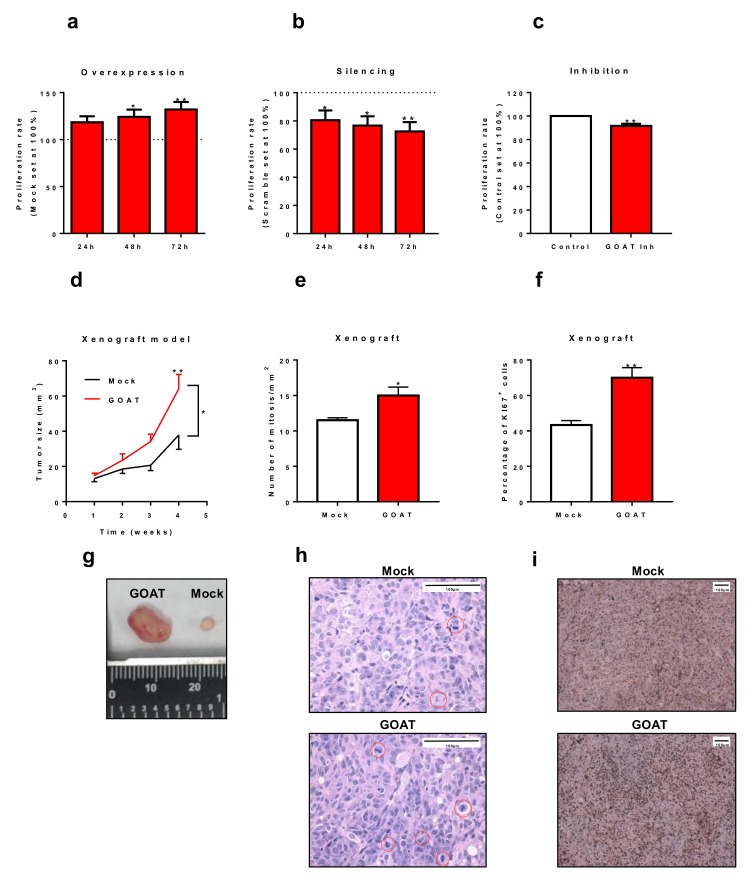
Effects of GOAT in vitro and in vivo. Cell proliferation rate (determined by Alamar-Blue assay) at 24, 48 and/or 72 h in response to *GOAT* overexpression (**a**), silencing (**b**) and pharmacological blockade with the specific GOAT inhibitor GO-CoA-Tat (**c**). Results are referred as a percentage of the control condition (mock, scramble and vehicle-treated cells, respectively). (**d**) Comparison between the growth of xenograft tumors derived from mock-transfected cells (black line) and GOAT-overexpressing cells (**red line**) over time. (**e**) Number of mitosis per mm^2^ in the xenograft tumors derived from mock-transfected DU145 cells and GOAT-overexpressing DU145 cells. (**f**) Percentage of positive KI67 cells in mock and GOAT xenograft tumors. (**g**) Representative image of tumor size of mock and GOAT xenograft tumors. (**h**) Representative image of xenograft tumors derived from mock-transfected DU145 cells and GOAT-overexpressing DU145 cells with hematoxylin-eosin staining. (**i**) Representative images of KI67 staining of xenograft tumors derived from mock-transfected DU145 cells and GOAT-overexpressing DU145 cells. The asterisks (*, *p* < 0.05; **, *p* < 0.01) indicate values that significantly differ between groups.

**Table 1 jcm-08-02056-t001:** Clinical and anatomopathological data of the three cohorts of patients included in this study.

Variable	Healthy *n* = 97 Cohort 1	NegBiopsy *n* = 549 Cohort 2	PCa *n* = 347		
			All *n* = 347 Cohort 3	NonSigPCa *n* = 143 Cohort 3a	SigPCa *n* = 204 Cohort 3b
Age Median (IQR)	62 (57–67)	63 (57–69)	67 (61–72)	65 (59–69)	69 (63–75)
PSA level (ng/mL) Median (IQR)	0.82 (0.57–1.33)	5.27 (3.84–7.39)	6.64 (4.49–11.32)	5.62 (3.79–9.09)	7.44 (4.83–16.09)
BMI	28.41 (25.54–32.09)	28.31 (25.96–30.86)	28.62 (26.30–31.63)	28.41 (26.44–31.15)	28.72 (26.15–32.05)
Patients with previous negative biopsy	-	174 (31.7)	66 (19.0)	35(24.5)	31 (15.2)
DRE (Abnormal)	-	59 (10.7)	125 (36.0)	34 (23.8)	91 (44.6)
5 alpha reductase inhibitors		20 (3.6)	5 (1.4)	1(0.7)	4 (0.2)
Family History		106 (19.3)	55 (15.9)	25 (17.5)	30 (14.7)
GS < 7	-	0	143 (41.2)		
GS ≥ 7	-	0	204 (58.8)		
Metastasis (%)	-	0	18 (5.2)	0	18 (8.8)

BMI: Body mass index; DRE: Digital rectal examination; PCa: Prostate cancer; SigPCa: Significant prostate cancer; IQR: Interquartile range; GS: Gleason score. Information about five alpha reductase inhibitors and family history of PCa was only collected for NegBiopsy and PCa patients.

**Table 2 jcm-08-02056-t002:** Multivariate analysis of the association of plasma GOAT levels with the diagnosis of prostate cancer in the full cohort of patients adjusting with common clinical variables.

Variable	OR	Bootstrap CI 95%	*p* Value
Age	1.036	1.014, 1.058	<0.001
PSA	1.047	1.019, 1.073	<0.001
Prior biopsy	0.522	0.3137, 0.7134	<0.001
GOAT	1.514	1.253, 1.755	<0.001
DRE	3.139	1.804, 4.307	<0.001

OR: Odds ratio; CI: Confidence interval; PSA: Prostatic specific antigen; DRE: Digital rectal examination.

**Table 3 jcm-08-02056-t003:** Multivariate analysis of the association of plasma GOAT levels with the diagnosis of Significant PCa (SigPCa) in the full cohort of patients adjusting with common clinical variables.

Variable	OR	Bootstrap CI 95%	*p* Value
Age	1.052	1.025, 1.079	<0.001
PSA	1.051	1.023, 1.075	<0.001
Prior biopsy	0.452	0.233, 0.662	<0.001
GOAT	1.477	1.171, 1.753	<0.001
DRE	2.878	1.639, 3.996	<0.001

OR: Odds ratio; CI: Confidence interval; PSA: Prostatic specific antigen; DRE: Digital rectal examination.

**Table 4 jcm-08-02056-t004:** Multivariate analysis of the association of plasma GOAT levels with the diagnosis of prostate cancer (PCa) in the patients with a PSA range of 3–10 ng/mL adjusting with common clinical variables.

Variable	OR	Bootstrap CI 95%	*p* Value
Age	1.039	1.013, 1.064	<0.001
PSA	1.144	1.014, 1.271	0.011
Prior biopsy	0.431	0.217, 0.633	<0.001
GOAT	1.553	1.226, 1.838	<0.001
DRE	1.553	1.212, 4.080	<0.001

OR: Odds ratio; CI: Confidence interval; PSA: Prostatic specific antigen; DRE: Digital rectal examination.

**Table 5 jcm-08-02056-t005:** Multivariate analysis of the association of plasma GOAT levels with the diagnosis of Significant PCa (SigPCa) in the patients with PSA a range of 3–10 ng/mL, adjusting with common clinical variables.

Variable	OR	Bootstrap CI 95%	*p* Value
Age	1.0532	1.020, 1.085	<0.001
PSA	1.119	0.958, 1.275	0.07
Prior biopsy	0.366	0.121, 0.596	0.001
GOAT	1.526	1.144, 1.859	<0.001
DRE	2.497	1.017, 3.762	<0.001

OR: Odds ratio; CI: Confidence interval; PSA: Prostatic specific antigen; DRE: Digital rectal examination.

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
