# Peer review of "Clinical Utility of Ghrelin-O-Acyltransferase (GOAT) Enzyme as a Diagnostic Tool and Potential Therapeutic Target in Prostate Cancer"

_jcm, 2019, doi:10.3390/jcm8122056_

Round 1
Reviewer 1 Report
The aim of the study was to analyze diagnostic value of GOAT in prostate cancer (PCa) and to show biological role of GOAT in PCa development.
Comments
1) Title should be revised according the aim of the study. There is no data in this study related with PCa treatment.
2) Experimental section:
Patients and samples: Is it a prospective or retrospective study? The number of patients who have had a previous negative biopsy is not clear in both cohorts 2 and 3. It should be stated and eventually analyzed separately. TRUS is no longer a standard and is replaced by MRI. It should be discussed. The start and end dates of inclusion should be added.
Elisa: The authors quantified GOAT with the same kit for urine and plasma. Urine was diluted 100 times before analysis. Did the authors control the matrix effect on GOAT quantification (linearity, LOD, reliability) in urine in comparison with plasma?
The kit manufacturer precise that serum should be used instead of plasma. (“It is highly recommended to use serum instead of plasma for the detection based on quantity of our in-house data”) . It could be a reason that results on plasma are poor as it was showed in the previous study published by the same team (quote#11). This should be discussed and controlled.
RT-PCR: It is well known that ACTB is not a stable gene in PCa. More generally, only one reference gene is not recommended to quantify truly mRNA. This should be corrected.
3) Results
There are confusions in legends between urine and plasma that should be corrected ( Fig 1 g-k).
"Fig 1e” is statistically significant. In the text (L229-230) it is stated that is not.
L230 Fig2d-f should be Fig1d-f.
ROC-curves analysis: 95%CI could be added.
Tables 2-4: It seems the tables are related to urine GOAT level instead GOAT plasma. Should be clarified
Tables 3-4: PSA should not be including in the multivariate analysis as it is used to select patients.
4) Discussion
The results obtained with GOAT should be compared with results obtained with many other well-studied biomarkers as PHI, MIPS, K4score, all biomarkers presenting Roc-curve AUC for early detection of Pca > 70% that is clearly superior to what achieved GOAT.
Reviewer 2 Report
The manuscript by Jiménez-Vacas, et al. aims to study the clinical utility of Ghrelin-O-Acyltransferase in urine samples of patients diagnosed with different stages of significance of prostate cancer and compared that with normal/healthy individuals. Although literature shows that the enzyme Ghrelin-O-Acyltransferase is overexpressed in prostate cancer and have been associated with higher potential value as a non-invasive biomarker, the current manuscript adds another layer of clarification to the ongoing interest in the field. Nevertheless, there are some flaws/missing points which are requested to be addressed by the authors.
Why did the authors choose to use only one prostate cancer cell line model? There is no data presented by the authors which shows the basal level of Ghrelin-O-Acyltransferase in different prostate cancer cell lines models as well as benign prostate cell lines. Further, DU145 represent a very aggressive set of prostate cancer cell line model. Keeping the above point in mind, is there any role of Ghrelin-O-Acyltransferase with AR? Is there any affect of Ghrelin-O-Acyltransferase on the microbiome of urine sample or vice-versa, especially focusing on the utility of Ghrelin-O-Acyltransferase as a non-invasive biomarker, what efforts were taken under consideration by the authors to rule this out. Are there any tissue(s) in the human body where Ghrelin-O-Acyltransferase is secreted or expressed without any disease setup (especially in the absence of prostate cancer)? PSA has been found to be non-specific in many cases and been shown to increase with mere inflammatory reaction, is this the same case for Ghrelin-O-Acyltransferase?
Reviewer 3 Report
The authors demonstrated plasma/urine Ghrelin O-Acyl Transferase enzyme (GOAT) levels were increased in prostate cancer patients. Moreover, they showed that urine GOAT levels were significantly increased in clinically significant prostate cancer patients.
They also presented that GOAT expression correlated with biological aggressiveness in prostate cancer in vivo and vitro.
However, there are lot of problems and questions that should be solved.
Major:
The authors just presented one point of GOAT levels of urine. Urine sometimes varies in creatinine concentrations. Therefore, the concentration of GOAT levels should be different. The authors should present some data with regards to the issue. Plasma levels of GOAT in this article were quite different compared to the previous report (Cancer letter 2016, ref 10). Please explain the differences. Why did the authors use DU145 as representative for prostate cancers? The reviewer requests to use other cell lines such as LNCaP and 22Rv1 to show GOAT biological aggressiveness. Total RNA from FFPE samples was isolated. How did the authors extract prostate cancer cells only? Prostate cancer cells are basically difficult to extract without contamination of interstitial cells. The authors should explain more precisely in Material and Methods. The reviewer recommends to present immunohistochemical experiments with regards to GOAT expression and CDK6, EGF, EZH2 and other molecular markers in human tissues. The authors should present GOAT expressions in DU145 cells overexpressing and knock-downed in Western blotting or qPCR. In Figure3 f, the representative photo of Ki67 expression is too exaggerated.
Minor:
In Figure 3 d the unit of X axis should be mm3 In Figure 1d, g, j X axis should be urine level of GOAT.
Round 2
Reviewer 1 Report
1) I would like to thanks the authors for their clear and relevant reply.
2)However, I think again that the title should be revised according the aim of the study.
The title " Clinical utility of Ghrelin-O-Acyltransferase (GOAT) enzyme as a diagnostic tool and therapeutic target in prostate cancer" is quite confusing because of the association of "clinical utility" and "therapeutic target".
There is no data in this study related with clinical utility of GOAT as therapeutic target. Goat is a potential therapeutic target as demonstrated by in vivo assays but was not tested in a clinical trial.
I strongly suggest to add ad minima:
Clinical utility of Ghrelin-O-Acyltransferase (GOAT) enzyme as a diagnostic tool and potential therapeutic target in prostate cancer"
I do not believe a paper deserve to have a title that could sound deceptive after reading the full paper but must instead reflect clearly the content of the study.
Author Response
Reviewer comment-1: I would like to thanks the authors for their clear and relevant reply.
Authors Response: Thank you for the laudatory comment of the Reviewer.
Reviewer comment-2: I think again that the title should be revised according the aim of the study.
Authors response: We sincerely thank the Reviewer for this suggestion. Therefore, following the Reviewer´s suggestion we have changed the title of the manuscript as follows: “Clinical utility of Ghrelin-O-Acyltransferase (GOAT) enzyme as a diagnostic tool and potential therapeutic target in prostate cancer”.
Reviewer 3 Report
The authors responded to the reviewer's comments appropriately.
Author Response
Reviewer comment: The authors responded to the reviewer's comments appropriately.
Authors Response: Thank you for the laudatory comment of the Reviewer.